# A Novel Propofol Dosing Regimen for Pediatric Sedation during Radiologic Tests

**DOI:** 10.3390/jcm11175076

**Published:** 2022-08-29

**Authors:** Ji-Young Min, Jeong-Rim Lee, Hye-Mi Lee, Ho-Jae Nam, Hyo-Jin Byon

**Affiliations:** 1Department of Anesthesiology and Pain Medicine, Eunpyeong St. Mary’s Hospital, College of Medicine, The Catholic University of Korea, Seoul 03312, Korea; 2Department of Anesthesiology and Pain Medicine, Anesthesia and Pain Research Institute, Yonsei University College of Medicine, 50-1 Yonsei-ro, Seodaemun-gu, Seoul 03722, Korea; 3Department of Anesthesiology and Pain Medicine, Anesthesia and Pain Research Institute, Yongin Severance Hospital, 363 Dongbaekjukjeon-daero, Giheung-gu, Yongin-si 16995, Korea

**Keywords:** children, propofol, radiological examination, sedation

## Abstract

The dose of propofol for pediatric sedation during radiologic tests has been proposed as an equation of 0.75 + 0.14 × age (months) + 45.82 × body surface area (m^2^) based on results in a previous study. We compared this equation and the conventional dosing strategy for sedation in children undergoing radiologic tests. An amount of 180 children scheduled for magnetic resonance imaging (MRI) were randomized to experimental and control groups. The initial induction dose of propofol calculated using the equation was administered in the experimental group. In the control group, children received 1 mg/kg of the initial induction dose of propofol. Then, 0.5 mg/kg of the additional dose was followed to induce sedation in both groups. When awake or moving, a rescue injection of 0.5 mg/kg propofol was given. The total induction dose was more significant in the experimental group. The number of injections for induction in the experimental group was lesser. The dose and number of rescue injections in the experimental group were significantly less. The equation for the induction dose of propofol in a previous study could achieve quick induction of sedation and prevent a rescue injection during sedation. However, caution is needed when using the equation.

## 1. Introduction

Sedation is required to facilitate various kinds of medical diagnoses or treatment procedures performed to reduce anxiety and secure the cooperation of patients. Children, especially, often need sedation because they have poor cooperation and a high-stress level [1,2]. Due to their anatomical and physical characteristics, children are vulnerable to side effects of sedatives, such as airway obstruction, apnea, and hypotension. For successful completion of procedures without the side effects of sedatives, an appropriate selection of sedatives and determination of dosage are crucial [3,4].

Radiologic tests such as computed tomography (CT) and magnetic resonance imaging (MRI) have characteristics different from other procedures for pediatric sedation. Children undergoing radiologic tests must be separated from their caregivers and stay in a machine for quite an extended period. The level of sedation for radiologic tests should be deep enough for children to keep motionless throughout tests to obtain accurate radiologic images [5]. Because radiologic tests do not cause pain, analgesics such as opioids are not required for sedation. Considering those characteristics of radiologic tests, the type and dosage of the sedative should be chosen [6,7,8].

Propofol has the advantages of rapid onset and recovery. It is widely used for sedation in children [9,10,11,12]. The dose of propofol has been proposed for pediatric sedation in previous studies [13,14]. However, the dose of propofol was presented without considering the characteristics of the children. In children, propofol’s pharmacokinetics and pharmacodynamics differ from those in adults. They are affected by the physical factors of being children, such as age and body surface area [15,16,17,18,19]. Furthermore, clinical research about the dose of propofol only for radiologic tests is rare.

We have performed regression analysis using medical records of successful pediatric sedation without the side effects of propofol for radiologic tests in our previous study [20]. The results of that study have shown that the induction dose of propofol is affected by age and body surface area of children. The induction dose (mg) of propofol was proposed as ‘0.75 + 0.14 × age (months) + 45.82 × body surface area (m^2^)’ based on a regression equation. Clinical studies are needed to evaluate the effect and safety of that equation. We hypothesized in the present study that the administration of doses calculated with formulas derived from the previous study [20] would result in fewer complications, the number of additional administrations, and doses. The present study aimed to compare the dose of propofol calculated with the equation proposed in the previous study with the conventional dose of propofol for sedation in children undergoing radiologic examinations.

## 2. Materials and Methods

### 2.1. Study Population

This prospective, randomized controlled study was performed in a single tertiary hospital. Informed consent was taken from all children’s parents after obtaining approval from the Institutional Review Board of Severance Hospital, Yonsei University Health System (approval number: 4-2017-0088). This study was registered at 26 August 2017 in http://cris.nih.go.kr (registration number: KCT 0002428)”. Children between 1 and 12 years of age with ASA physical status I or II who underwent sedation for an MRI scan with an expected scan duration of <30 min were enrolled in this study. Patients with known respiratory or cardiac disease, neurologic deficits, upper respiratory infection, an anomaly of the airway, and those who received analgesics or sedatives within the previous 24 h were excluded from this study.

### 2.2. Pediatric Sedation Protocol

No premedication was administered before the imaging test. Before arrival in the MRI room, a 24-gauge cannula was inserted into the dorsum of the hand. Dextrose or saline was connected. Upon arrival in the MRI room, all children were monitored with an electrocardiogram, a pulse oximeter, a non-invasive blood pressure device, and capnography. Vital signs, including non-invasive blood pressure, heart rate, and end-tidal CO_2_, were recorded for five minutes throughout the study. They were randomly assigned to the experimental and control groups according to a computerized, randomized table. Allocations were concealed in sequentially numbered, sealed, opaque envelopes. When the patient entered the room, oxygen was administered at 3–5 L through the facial mask. For children assigned to the experimental group, the initial induction dose of propofol (mg) was calculated using the equation of 0.75 + 0.14 × age (months) + 45.82 × body surface area (m^2^). Body surface area was defined as (height (cm) × weight (kg)/3600)½. After the initial induction dose of propofol, 0.5 mg/kg of an additional induction dose was followed at 30 s intervals until a Ramsay sedation score of 4–5 was achieved. Once the Ramsay sedation score reached 4–5, an MRI scan was started. Anesthetic dosing was limited to propofol boluses, and no infusions were used for both groups. If there was a movement of the children after the MRI scan was started, a rescue injection of propofol at 0.5 mg/kg was given at 30 s intervals until a Ramsay sedation score of 4–5 was achieved. The initial induction dose of propofol for children assigned to the control group was 1 mg/kg according to the Korean Pediatric Sedation Guideline (https://pedianesth.or.kr/board/list.html?num=1180&code=docu03, accessed on 26 August 2017). The process after injection of the initial induction dose was the same as in the experimental group. The time propofol was injected and the dose of propofol was recorded during the study period. The total induction dose was defined as the dose of propofol injected to achieve a Ramsay sedation score of 4–5 at the beginning of sedation. Except for an independent researcher who prepared and administered the initial induction dose of propofol, other sedation providers and children were blinded to the group the patient was assigned to. Side effects of propofol included hypotension, bradycardia, and arterial desaturation. Hypotension and bradycardia were defined as a decrease of 30% or more from the initially measured value. Arterial desaturation was defined when SpO_2_ was less than 95%. After the MRI scan ended, children were sent to the recovery room and monitored with an electrocardiogram, a pulse oximeter, a non-invasive blood pressure device, and capnography. Recovery time was assessed as the time from entering the recovery room until children showed a Ramsay sedation score of 1 or 2. Children were discharged when they showed an Aldrete score of >8 points.

### 2.3. Statistical Analysis

The primary outcome was to evaluate the incidence of complications (%) between the two groups. Considering that the incidence of respiratory side effects was 23% after the use of propofol in a previous study [20], the sample size was determined by assuming that complications would be decreased by 15% when propofol was administered using the propofol dosage formula proposed in the previous study. We calculated that 90 patients would be needed for each group (experimental group and control group) to achieve an α = 0.05 and a power of 0.80 with a dropout from a study of 10%. Thus, the total sample size was 180 patients. The secondary outcome was defined as the difference in the total dose of propofol, the number of times added during the examination, and the difference in additional propofol dose. Statistical analyses were performed using IBM SPSS Statistics, version 25.0 (IBM Corp., Armonk, NY, USA). The normality of data was tested by the Shapiro–Wilk test and the Kolmogorov–Smirnov test. All data are expressed as a number (%) or median (IQR). Data were compared between the two groups using the Chi-square test, Fisher’s exact test, or the Mann–Whitney U-test as appropriate. The Bonferroni correction performed multiple testing corrections. Statistical significance was defined at *p* < 0.05.

## 3. Results

A total of 180 children undergoing MRI scans were enrolled in this study. Two children in the control group were excluded from statistical analysis because of incidental obstruction and removal of intravenous catheters. (Figure 1.) There was no significant difference in demographic characteristics of children such as age, sex, height, weight, or × body surface area (m^2^) between the two groups (Table 1). Underling disorders of children for MRI scans are shown in Table 2, with neurologic disorders being the most common. In all children, a Ramsay’s sedation score of 4–5 was achieved and maintained throughout the MRI scan. The total induction dose in the experimental group was significantly higher than that in the control group (40.5 (30.8–51.3) mg vs. 34.0 (27.5–45.0) mg, *p*-value = 0.011). However, the total propofol dose mg/kg in the experimental group was significantly lower than that in the control group (2.8 (2.4–3.4)) mg/kg vs. 3.0 (2.0–3.0) mg/kg, *p*-value = 0.001) (Table 3). However, the rescue dose was significantly lower in the experimental group than in the control group (0.0 (0.0–17.3) mg vs. 14.0 (0.0–24.5) mg, *p*-value = 0.032). The total dose administered during the MRI scan was not significantly different between the two groups. The number of injections for induction was significantly lower in the experimental group than in the control group (2.0 (1.0–2.0) vs. 3.0 (2.0–3.0), *p*-value = 0.0001). The number of rescue injections in the experimental group was also significantly lower than that in the control group (0.0 (0.0–1.0) vs. 1.0 (0.0–2.0), *p*-value = 0.005). The radiologic test’s duration of sedation, recovery time, and side effects of propofol were not significantly different between the two groups. Recovery time was less than 30 min in all children except for one child in the experimental group and three in the control group. Arterial desaturation occurred in one child of the experimental group due to airway obstruction and two children of the control group due to aspiration and airway obstruction. All children experiencing arterial desaturation recovered without interrupting the MRI scans.

## 4. Discussion

In this randomized prospective study, we compared the initial induction dose of propofol calculated by the equation of 0.75 + 0.14 × age (months) + 45.82 × body surface area (m^2^) proposed in our previous study with the conventional induction dose (1 mg/kg) for pediatric sedation during radiologic tests [12,21]. The total induction dose was more significant when the initial induction dose was calculated using the regression equation proposed in our previous study compared to the conventional induction dose. In contrast, the number of injections for induction, rescue dose, and the number of rescue injections were reduced in the experimental group. However, the total dose of propofol, recovery time, and side effects were not significantly different between the experimental group (initial induction dose was administered based on the equation) and the control group (initial induction dose was the conventional induction dose).

The determination of the proper sedative dose of propofol is essential to achieve sedation without complications during radiologic tests in children. Too high a dose of propofol can lead to complications such as hypotension, airway obstruction, and apnea. A too low dose of propofol cannot achieve enough of a level of sedation for radiologic tests. Thus, calculating the amount of propofol for pediatric sedation can be a challenge for clinicians [22,23]. The equation calculating the initial induction dose of propofol used in the present study has been proposed in our previous survey after performing a retrospective regression analysis of doses of propofol administered for CT or MRI scan [20]. However, clinical research studies have not been performed to verify that equation’s effectiveness or safety.

In the experimental group, the total induction dose was significantly higher, and when analyzed per weight, the experimental group had a statistically lower dose than the control group. The number of additional injections for induction was fewer than those in the control group. The total induction dose used in this study was a combination of the initial induction dose calculated by the equation and the additional dose administered until a Ramsay sedation score of 4–5 was achieved. Compared to the control group, the initial dose calculated using the equation in the experimental group was more significant enough to induce sedation and reduce additional administration for the induction of sedation. In other words, more than 50% of experimental group subjects were sedated with the dose calculated by the equation, and the imaging test could be started. This indicates that the equation can guide physicians in determining the induction dose of propofol to achieve the proper level of sedation for radiologic tests more appropriately than when using a dose calculated by the weight.

The equation used in the experimental group yielded an initial induction dose of propofol for pediatric sedation based on age and BSA. It is known that there is a difference in dose depending on age. However, few studies have calculated the dose by substituting age as a number like weight [24]. BSA is the external surface of the body expressed in square meters (m^2^). It represents the relationship between height and weight, thus providing a more accurate guide to the maturity of body organs and metabolic rate, which are essential factors in drug absorption, distribution, metabolism, and excretion. Using the BSA to determine drug dosages may allow physicians to select doses suitable for individual characteristics. The experimental group’s initial induction dose was chosen based on the children’s age, height, and weight. However, the conventional induction dose for the control group was based solely on the children’s weight. The induction dose calculated by the equation reflected various characteristics of children. This might be one of the reasons why the number of injections for induction was decreased when the equation was used to calculate the initial induction dose.

The rescue dose and the number of rescue injections were less when the initial induction dose calculated by the equation was used than when the conventional induction dose was used. The initial induction dose calculated by the equation can reduce the number of rescue injections of propofol to maintain the level of sedation during radiologic tests. It can prevent the possibility of interruption of radiologic tests due to the consciousness or movement of children, meaning that more accurate images can be obtained continuously. Imaging tests are necessary to diagnose and determine the severity of a disease. In the case of MRI, the accuracy of the image is poor, even with the slightest movement of the patient. Pictures can be taken without interruption and provide accurate information about the patient’s current state. Physicians also can save their time and efforts by injecting propofol. This was because the initial induction dose of propofol calculated by the equation was more significant than the conventional dose. When a large amount of propofol is injected, the effect of propofol can last longer. In this study, there was no propofol infusion because MRI scans lasted about 25 min. If the duration of radiologic tests was longer or if propofol was infused, the result of rescue injections could be different.

This study’s recovery time and side effects of propofol were not significantly different between the experimental and control groups. This was because the total dose of propofol was not significantly different between the two groups. Incidences of adverse events were lower than those in previous studies [20]. Except for four children in the two groups, all children recovered from the sedation within 30 min. There were no side effects other than transient arterial desaturation in three children in the two groups. In the control group, it is possible that because 1 mg/kg of the initial induction dose was less than that in previous studies, the initial induction dose was 1–2 mg/kg [25,26]. In the experimental group, it might be because the equation calculating the initial induction dose was based on data on the induction dose from successful pediatric sedation cases without complications. The formula used in this study presents larger induction doses than the 1 mg/kg suggested in the existing guideline. The conventional approaches require stopping the examination and additional administration with a small dose. With the formula in the present study, sedation succeeds with a small number of administrations. Side effects from large doses also did not occur. However, since a large induction dose of propofol calculated by the equation is administered, physicians should be careful when sedating children vulnerable to propofol’s side effects.

This study was conducted on children undergoing MRI scans. Characteristics of sedation for MRI scans can affect the results of the investigation. An MRI scan does not induce pain. Sedation for MRI scans does not need an analgesic effect. To obtain accurate images, children should keep motionless during MRI scans. Even a tiny movement can interfere with the MRI test. However, children do not need to cooperate with a physician. Therefore, applying the results of this study to children undergoing sedation for other procedures or examinations with different characteristics requires caution.

This study has several limitations. First, this study enrolled children aged between 1 and 12 years. Results of this study should only be applied to children aged one year or older. Further research is needed on children under the age of one. Second, this study excluded children with respiratory or cardiac disease, neurologic deficits, upper respiratory infection, or airway anomaly. These children can be vulnerable to the side effects of propofol. The results of this study may be different for these children. Thus, physicians should be careful when applying the results of this study to these children excluded from this study. Third, we used the Ramsay sedation score to monitor the level of sedation. There was no objective device for evaluating the level of sedation, such as an EEG monitor in an MRI room.

## 5. Conclusions

In children undergoing sedation for an MRI scan, the initial induction dose of propofol calculated by the equation of 0.75 + 0.14 × age (months) + 45.82 × body surface area (m^2^) can decrease the number of injections for induction and rescue doses. The calculated dose by formula can reduce the number of injections compared to a standard 1 mg/kg weight-based dosing. Compared to the conventional method, this advantage could allow the radiologic examination to continue without interruption, delaying recovery, or causing side effects. However, due to a sizeable total induction dose, the physician should be careful when using the equation to calculate the induction dose for pediatric sedation.

## Figures and Tables

**Figure 1 jcm-11-05076-f001:**
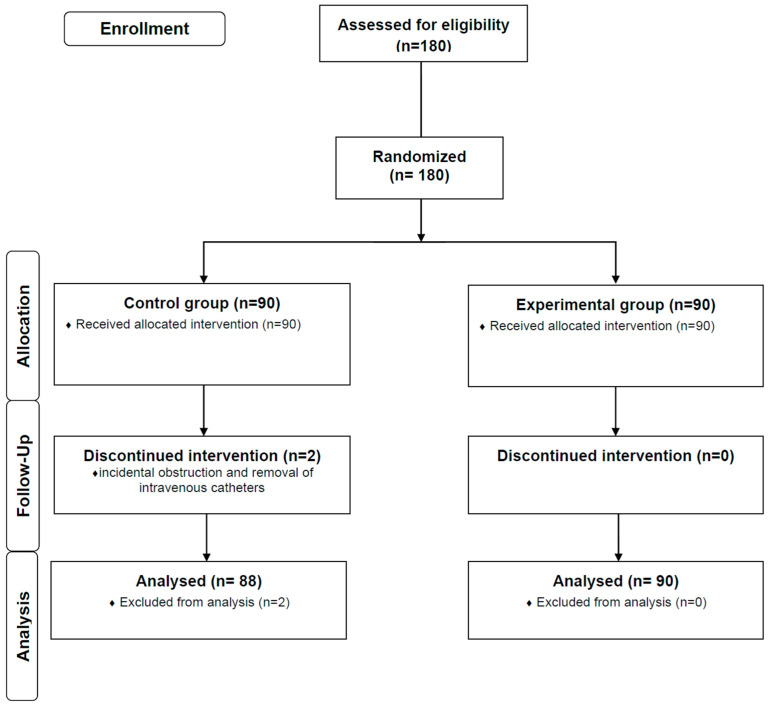
Consort flow diagram.

**Table 1 jcm-11-05076-t001:** Demographic characteristics of children undergoing sedation for MRI scan.

	Control Group (*n* = 88)	Experimental Group (*n* = 90)	*p*-Value
Age (month)	31.0 (21.0–49.0)	39.5 (25.0–51.5)	0.121
Sex (male/female)	54/34	55/35	0.972
Weight (kg)	13.8 (10.4–16.4)	14.6 (11.9–16.0)	0.429
Height (cm)	90.0 (80.0–99.9)	95.0 (83.0–102.0)	0.118
BSA (m^2^)	0.58 (0.48–0.67)	0.61 (0.52–0.68)	0.215

Values are presented as a number (%) or median (IQR). BSA, body surface area (m^2^).

**Table 2 jcm-11-05076-t002:** Underlying diseases of children undergoing sedation for MRI scan.

	Control Group (*n* = 88)	Experimental Group (*n* = 90)	*p*-Value
Neurologic disorder	57 (64.8%)	55 (61.1%)	0.613
Seizure disorder	20	16	
Anatomical malformation	16	22	
Tumor	11	10	
Vascular disease	6	2	
Shunt related problem	2	-	
Trauma	2	5	
Genetic disorder	15 (17.0%)	9 (10.0%)	0.169
Delayed development	11	9	
Anorectal anomaly	2	-	
Down syndrome	2	-	
Ophthalmologic disorder (Tumor or mass)	4 (4.5%)	6 (6.7%)	0.747 *
Orthopedic disease (Fracture or malformation)	2 (2.5%)	5 (5.6%)	0.444 *
Others	10 (11.4%)	15 (16.7%)	0.309

Values are presented as a number (%). * The *p*-value was calculated by Fisher’s exact test.

**Table 3 jcm-11-05076-t003:** Outcome measures of children undergoing sedation during the radiologic examination.

	Control Group (*n* = 88)	Experimental Group (*n* = 90)	*p*-Value
Total induction dose (mg)	34.0 (27.5–45.0)	40.5 (30.8–51.3)	0.011
Total induction dose (mg/kg)	3.0 (2.0–3.0)	2.8 (2.4–3.4)	0.001
Number of injections for induction	3.0 (2.0–3.0)	2.0 (1.0–2.0)	0.0001
Rescue dose (mg)	14.0 (0.0–24.5)	0.0 (0.0–17.3)	0.032
Number of rescue injections	1.0 (0.0–2.0)	0.0 (0.0–1.0)	0.005
Total dose of propofol (mg)	47.0 (33.5–64.5)	49.5 (34.8–66.5)	0.823
Duration of radiologic test (minutes)	25.0 (20.0–31.0)	25.0 (20.0–31.0)	0.908
Duration of sedation (minutes)	33.0 (27.0–43.0)	35.0 (28.8–45.0)	0.450
Recovery time (minutes)	9.0 (1.0–15.0)	10.0 (2.0–17.0)	0.349
Side effects of propofol	2/86	1/89	0.619

Values are presented as a median (IQR) or a number (%).

## Data Availability

The datasets used and analyzed during the current study are available from the corresponding author upon reasonable request.

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
