# Peer review of "A Novel Propofol Dosing Regimen for Pediatric Sedation during Radiologic Tests"

_jcm, 2022, doi:10.3390/jcm11175076_

Round 1

Reviewer 1 Report

The goal of this study was to compare the dose of propofol calculated with the equation proposed in the previous study with the conventional dose of propofol for sedation in children under- going radiologic examinations. 

Methods: Pre-determined primary and secondary outcomes of the study and their measures might be described in more details.

Results:

  Table 2.Underlying diseases of children undergoing sedation for MRI scan.   Table 2 contains neurological, generic disorders and etc. Since the current titles of these disorders are not specific and can vary in severity, the specification ( type of neurologic. genetic, ophthalmologic disorders)  might be needed.  

Table 3. Doses are given as "total in mg". Since the body weight varies widely in pediatric patients, would doses given in "mg per kg of the body weight" be more precise?

Can this novel anesthetic dosing regiment reduce the dose-related complications of the standard dosing regiment and improve patients' outcomes?

Were there any unintended harms to the participants during the study?

Author Response

Thank you for your constructive comments. It helps us considerably improve the manuscript and enhance its clarity. We have responded to each of your comments point-by-point with the changes marked in red. In addition, we have provided a revised version of the manuscript with the changes. We hope that the changes made to the revised manuscript satisfactorily address your concerns.

Reviewer 1.

The goal of this study was to compare the dose of propofol calculated with the equation proposed in the previous study with the conventional dose of propofol for sedation in children under- going radiologic examinations.

Methods: Pre-determined primary and secondary outcomes of the study and their measures might be described in more details.

  • We added additional comments in the “statistical analysis” section.
  • The primary outcome was to evaluate the incidence of complications (%) between the two groups. Considering that the incidence of respiratory side effects was 23% after using propofol in a previous study, the sample size was determined by assuming that complications would be decreased by 15% when propofol was administered using the propofol dosage formula proposed in the previous study. The secondary outcome was defined as the difference in the total dose of propofol, the number of times added during the examination, and the difference in additional propofol dose.

Results:

Table 2.Underlying diseases of children undergoing sedation for MRI scan.  

Table 2 contains neurological, generic disorders and etc. Since the current titles of these disorders are not specific and can vary in severity, the specification ( type of neurologic. genetic, ophthalmologic disorders)  might be needed.

  • As you recommended, we newly added the subcategories of disease in Table 2.

Table 3. Doses are given as "total in mg". Since the body weight varies widely in pediatric patients, would doses given in "mg per kg of the body weight" be more precise?

  • As you mentioned, the analysis result corresponding to mg/kg is inserted in Table 3. In practice, dose per weight was analyzed, and the experimental group showed lower values per kg.

Can this novel anesthetic dosing regiment reduce the dose-related complications of the standard dosing regiment and improve patients' outcomes?

  • As a result of the study, the difference in complications between the two groups was insignificant. (p=0.619) There is no difference in complications, but the experimental group can complete the test at once without complications despite of higher dose, because of the number of times the examination is discontinued compared to the control group. This is thought to reduce the administration of unnecessary drugs in patients and reduce the effect of accumulation by additional doses, thereby reducing complications that may occur during recovery in clinical practice.

Were there any unintended harms to the participants during the study?

  • There was no unintended harm to the participants during the study.

Reviewer 2 Report

The authors present the implementation of a novel formula for induction propofol dosing based on a prior study, comparing it to a standard weight-based dosing induction. The authors should be commended on an excellent job with study design, including randomization and blinding.

One area of concern is the potential for bias due to the low weight-based dose used for induction, and it would be helpful to know how the dose of 1 mg/kg was chosen and why groups of, say, 2 mg/kg or 3 mg/kg were not also included. The weight-based dosing choice creates a situation where the experimental group almost always received a much higher first dose of propofol – for example, an average child in the control group that is 31 months and weighs 13.8 kg and has a BSA of 0.58 m^2 would be over 31 mg, almost 2.5 mg/kg. This is reflected in the fact that the experimental group had significantly higher total induction doses despite a fewer number of injections. In light of this, it is difficult to know whether the results reflect the effectiveness of the formula or whether a higher mg/kg dosing is more appropriate for induction for MRI.

Specific comments:

Intro: was there a hypothesis for the study?

Methods: scan duration < 20 minutes? Which scans are these?

L79: change arterial blood pressure to non-invasive blood pressure

L84: should state that this induction dose is in milligrams

L85: please state the formula used to calculate body surface area

L86: how was anesthesia maintained after the initial injections? Was an infusion started, or were only rescue doses given for the entire 25-minute MRI?

L91: please state how the dose of 1 mg/kg was chosen

L109: was this a formal power analysis? If so, would change ‘estimated’ to ‘calculated’.

Table 3: I found it interesting that the control group had slightly shorter anesthetics despite requiring more injections to get started and more rescue injections. Any idea why this might be?

L217: it cannot be assumed that the reduced dose fully explains the lack of events, please change to ‘it is possible that because 1 mg/kg of in the initial…’

L246: please clarify that the calculated dose can reduce the number of injections compared to a standard 1 mg/kg weight-based dosing

Author Response

Thank you for your constructive comments. It helps us considerably improve the manuscript and enhance its clarity. We have responded to each of your comments point-by-point with the changes marked in red. In addition, we have provided a revised version of the manuscript with the changes. We hope that the changes made to the revised manuscript satisfactorily address your concerns.

Reviewer 2

  • The authors present the implementation of a novel formula for induction propofol dosing based on a prior study, comparing it to a standard weight-based dosing induction. The authors should be commended on an excellent job with study design, including randomization and blinding.
  • I want to express my sincere appreciation for your comment on this study.
  •  
  • One area of concern is the potential for bias due to the low weight-based dose used for induction, and it would be helpful to know how the dose of 1 mg/kg was chosen and why groups of, say, 2 mg/kg or 3 mg/kg were not also included. The weight-based dosing choice creates a situation where the experimental group almost always received a much higher first dose of propofol – for example, an average child in the control group that is 31 months and weighs 13.8 kg and has a BSA of 0.58 m^2 would be over 31 mg, almost 2.5 mg/kg. This is reflected in the fact that the experimental group had significantly higher total induction doses despite a fewer number of injections. In light of this, it is difficult to know whether the results reflect the effectiveness of the formula or whether a higher mg/kg dosing is more appropriate for induction for MRI.
  • Thank you for your precise point. We are also concerned about the potential for the problem you asked about. The formula used in this study presents larger induction doses than the 1 mg/kg suggested in the existing guideline. Existing guidelines require additional administration with a small dose, but with our formula, sedation succeeds with a small number of additional administrations. Side effects from large doses also did not occur. In addition to this, our new formula allows to consider the factors in addition to weight. For example, they assume that there are 50-month-old children who are 110cm tall and 27kg and 100-month-old children who are 137cm tall and 27kg. 49mg and 61mg are calculated, respectively, according to the formula used in this study. When calculated by weight, both children will be given the same dose.
  • Unlike previous studies, the dose in the present study was calculated through a formula that reflects both pharmacological and pharmacokinetic characteristics of children and their unique characteristics as independent individuals, not a miniature of an adult. Unlike high-dose propofol based on weight, it can be considered in dose much closer to children's specific characteristics. The same high dose may be used but our calculated drug dose echoed more detailed features of pediatric patients.

Specific comments:

Intro: was there a hypothesis for the study?

  • We hypothesized in the present study that administration of doses calculated with formulas derived from previous research would result in fewer complications, the number of additional administrations, and fewer doses. We amended the last section of introduction.

Methods: scan duration < 20 minutes? Which scans are these?

  • We corrected the typo. We changed “30 minutes.”

L79: change arterial blood pressure to non-invasive blood pressure

  • We changed that phrase.

L84: should state that this induction dose is in milligrams

  • We inserted the (mg) behind “propofol’.

L85: please state the formula used to calculate body surface area

  • We inserted that “Body surface area was defined as (height (cm) x weight (kg)/3600)½.”.

L86: how was anesthesia maintained after the initial injections? Was an infusion started, or were only rescue doses given for the entire 25-minute MRI?

  • We did not offer infusion for both groups but only bolus injection. We inserted that “We did not infusion for both groups”.

L91: please state how the dose of 1 mg/kg was chosen

  • The initial induction dose of propofol at 1 mg/kg was injected for children assigned to the control group was 1 mg/kg according to the Korean Pediatric Sedation Guidelines. As you recommended, we mentioned how the dose of 1 mg/kg was chosen.

L109: was this a formal power analysis? If so, would change ‘estimated’ to ‘calculated.’

  • We change ‘estimated’ to ‘calculated.’

Table 3: I found it interesting that the control group had slightly shorter anesthetics despite requiring more injections to get started and more rescue injections. Any idea why this might be?

  • Statistically, there is no difference between the two groups. This result may be that most of the additional dose was given at the beginning of the test rather than at the end of the examination.

L217: it cannot be assumed that the reduced dose fully explains the lack of events, please change to ‘it is possible that because 1 mg/kg of in the initial…’

  • We inserted “it is possible that.”

L246: please clarify that the calculated dose can reduce the number of injections compared to a standard 1 mg/kg weight-based dosing

  • In the conclusion section, we have clarified the advantages of the present study.

Round 2

Reviewer 2 Report

Thank you for addressing my points. I would suggest two further minor changes:

Line 92: change the added line to 'Anesthetic dosing was limited to propofol boluses and no infusions were used.'

Line 96: Please provide a citation or link to the Korean sedation guidelines.

Author Response

Thank you for your constructive comments. It helps us considerably improve the manuscript and enhance its clarity. We have responded to each of your comments point-by-point with the changes marked in red. In addition, we have provided a revised version of the manuscript with the changes. We hope that the changes made to the revised manuscript satisfactorily address your concerns.

Line 92: change the added line to 'Anesthetic dosing was limited to propofol boluses and no infusions were used.'

We changed the add line as you recommended.

Line 96: Please provide a citation or link to the Korean sedation guidelines.

We have added a link to the guidelines mentioned in the paper.

In the present study, we considered the results of references 25 and 26 as well as the corresponding guidelines.
